# Reproducibility review of *"Why Not Other Classes?": Towards Class-Contrastive Back-Propagation Explanations*

## Abstract

*"Why Not Other Classes?": Towards Class-Contrastive Back-Propagation Explanations* (Wang & Wang, 2022) provides a method for contrastively explaining why a certain class in a neural network image classifier is chosen above others. This method consists of using back-propagation-based explanation methods from after the softmax layer rather than before. Our work consists of reproducing the work in the original paper. We also provide extensions to the paper by evaluating the method on XGradCAM, FullGrad, and Vision Transformers to evaluate its generalization capabilities. The reproductions show similar results as the original paper, with the only difference being the visualization of heatmaps which could not be reproduced to look similar. The generalization seems to be generally good, with implementations working for Vision Transformers and alternative back-propagation methods. We also show that the original paper suffers from issues such as a lack of detail in the method and an erroneous equation which makes reproducibility difficult. To remedy this we provide an open-source repository containing all code used for this project.

## 1 Introduction

Deep Neural Networks (DNNs) have seen rapid growth in recent years due to their great performance across many fields. However, these high-performing models suffer from being black-box, and therefore are hard to interpret the decision-making process of. This is especially dangerous in security-critical systems, such as in medicine or autonomous driving, where full transparency of decisions is needed. As an approach to making DNNs more interpretable the field of Explainable AI studies different methods for explaining the decision process of DNNs. A paper that studies such an approach, with a focus on computer vision and back-propagation-based methods is the paper under scrutiny in this review.

The original paper *"Why Not Other Classes?": Towards Class-Contrastive Back-Propagation Explanations* (Wang & Wang, 2022) propose a new weighted contrastive back-propagation-based explanation. This method aims to improve the explanation of why one specific class are chosen over others. By answering the question of what differs between two similar classes, rather than what is important for both, the goal is to get a explanation method that closer matches how people answers classification tasks.

Their proposed explanation method, called weighted contrast, are a class-wise weighted combination of the original explanation defined as

$$\phi_i^t(\boldsymbol{x})_{\text{weighted}} = \phi_i^t(\boldsymbol{x}) - \sum_{s \neq t} \alpha_s \phi_i^s(\boldsymbol{x}) \tag{1}$$

where $\phi_i$ is the original explanation for pixel $i$ and the weight $\alpha$ is the softmax activation of the logit vector without the target class $t$

$$\alpha_s = \frac{\exp y_s}{\sum_{k \neq t} \exp y_k} \tag{2}$$

The original explanation can be any back-Propagation based explanation method. This paper will further investigate three of the methods proposed, namely, Gradient, Linear Approximation (LA) and GradCAM as detailed in Appendix A. The original paper further shows that the weighted contrast method is equal to taking the explanation directly toward the probability after the softmax layer for most gradient-based methods.

The authors argue that this is a superior contrastive explanation method by performing two forms of adversarial attacks with regard to the different explanations. They show that an adversarial attack on the pixels highlighted by weighted contrast results in a more significant effect on the accuracy of the model, while original methods more accurately impact the logit strength. By performing a blurring and removal attack with explanations extracted from GradCAM and Linear Approximation they show that their method finds more impactful negative and positive regions of interest with regards to the model accuracy.

This document aims to reproduce the main results of the paper as well as provide insights into the general reproducibility and impact of the paper. We also expand upon the paper and attempt applications outside its scope, with other back-propagation-based explanation methods as well as applying it to Vision Transformers (ViT) as introduced in Dosovitskiy et al. (2020). This was done to see the generalization capabilities of the paper.

## 2   Scope of reproducibility

The claims of the original paper we seek to verify are:

- **Claim 1:** When perturbing input features according to the original paper's weighted contrastive explanation, changes in the softmax output $p_t$ and accuracy match each other, whilst changes in the logit $y_t$ do not for target class $t$.

- **Claim 2:** The original paper's weighted contrastive explanation when coupled with a visualization method such as GradCAM highlights the most crucial areas for classification when the model classifies between several dominant classes.

- **Claim 3:** The original paper's weighted contrastive explanation should be able to be easily applied to other back-propagation-based explanation methods by back-propagating from the softmax output $p_t$ rather than the logit $y_t$.

In order to evaluate the above claims and thus test the reproducibility of the paper we replicated the steps described in section *5 Experiments* of the original paper for a subset of the models and datasets used in the original paper. We thus made our own experiments using our own code. The results of these experiments were then compared with those of the paper in order to find if they were consistent. We furthermore test the generalizability of the paper by applying the contrastive explanation method shown in the original paper using XGradCAM (Fu et al., 2020), FullGrad (Srinivas & Fleuret, 2019), and Vision Transformers (Dosovitskiy et al., 2020).

### 2.1   XGradCAM and FullGrad

As an attempt to test the paper's contrastive method's generalization capability additional back-propagation methods in the form of modified versions of XGradCAM (Fu et al., 2020) and FullGrad (Srinivas & Fleuret, 2019) were used.

The modified version of XGradCAM removes ReLU, as in the original paper, and as a consequence uses the absolute values in the feature map sum when normalizing rather than only the sum. This gives the following explanation $\boldsymbol{\phi}^t(\boldsymbol{x})_y$ when back-propagating from the logit $y_t$ with the target feature map layer $\boldsymbol{a}$ with $k \in K$ feature maps:

$$\boldsymbol{\phi}^t(\boldsymbol{x})_y = \sum_k \left( \sum_{i,j} \frac{a_{ij}^k}{\|\boldsymbol{a}^k\|_1} \frac{\partial y_t}{\partial a_{ij}^k} \right) \boldsymbol{a}^k \tag{3}$$

A weighted contrastive version, $\boldsymbol{\phi}^t(\boldsymbol{x})_{\text{weighted}}$ as described in the original paper, of XGradCAM can be obtained by propagating from the softmax neuron $p_t$ and can be proven as follows using notation $[c]$ for all classes:

$$\boldsymbol{\phi}^t(\boldsymbol{x})_p = \sum_k \left( \sum_{i,j} \frac{a_{ij}^k}{\|\boldsymbol{a}^k\|_1} \sum_{s\in[c]} \left( \frac{\partial p_t}{\partial y_s} \frac{\partial y_s}{\partial a_{ij}^k} \right) \right) \boldsymbol{a}^k = \sum_{s\in[c]} \frac{\partial p_t}{\partial y_s} \boldsymbol{\phi}^s(\boldsymbol{x})_y \propto \boldsymbol{\phi}^t(\boldsymbol{x})_{\text{weighted}} \tag{4}$$

The modified version of FullGrad changes the post-processing operations indicated by $\psi$ in (Srinivas & Fleuret, 2019) by removing the abs function in order to be linear and therefore allow negative and positive values in the produced saliency map. This allows us to generate a contrastive weighted version by back-propagating from the target softmax neuron $p_t$ in but also heavily alters the method.

## 2.2 Vision Transformers

In order to test how differences in architecture affect the results we modified two sets of explanation methods, GradCAM and and Gradient-weighted attention rollout (Gildenblat, 2020), and tested them together with the `vit_b_16` model as first described in Dosovitskiy et al. (2020). This model works by dividing the image into 16x16 patches, interpreting each patch of the image as a token. The information in these layers is then propagated throughout the network using a self-attention mechanism. Unlike standard convolutional neural network (CNN) architectures, spatial coherence is not guaranteed through the network, and information is easily mixed with some layers containing little to no spatial coherence.

## 3 Experiments

In this section, we detail the various reproduction experiments and additions to the original paper. They were performed using the `PyTorch` library and the code is available publicly at https://anonymous.4open.science/r/contrastive-explanations-58EE/ under the MIT License. All experiments were performed on a `n2-standard-4` Google Cloud VM with an NVIDIA T4 GPU.

### 3.1 Reproducing 5.1 Back-Propagation till the Input Space

This section reproduces the experiments from section 5.1 in the original paper. The experiments test nine networks with perturbed input images where the perturbation uses four different explanation methods to select pixels to perturb. The four methods are original, mean, max and weighted.

*Original* is gradient explanation defined as $\phi^t(\mathbf{x}) = \nabla_{\mathbf{x}} y^t$.

*Mean* is the original explanation averaged over all classes as, $\phi^t(\mathbf{x}) = \nabla_{\mathbf{x}} y^t - \sum_{s\neq t} \nabla_{\mathbf{x}} y^s$.

*Max* is considering only the correct class and the highest other class, defined as $\phi^t(\mathbf{x}) = \nabla_{\mathbf{x}} y^t - \nabla_{\mathbf{x}} y^{s^*}$, where $s^* = arg\max_{s\neq t} y^s$

*Weighted* is the original papers new method shown in (1), using the *original* explanation method, which gives $\phi^t(\mathbf{x}) = \nabla_{\mathbf{x}} y^t - \sum_{s\neq t} \alpha_s \nabla_{\mathbf{x}} y^s$, where $\alpha$ is given by (2).

All models use `PyTorch` pre-trained models, with the most up-to-date default weights as of writing, and are tested on the validation set of ILSVRC2012 (Deng et al., 2009). The experiments are repeated with a perturbation limit, $\epsilon$, of $3 \times 10^{-3}$, see Figure 1. This differs from the original papers reported $\epsilon = 10^{-3}$, while after being in contact with the original authors we found that $\epsilon = 3 \times 10^{-3}$ had been used. An experiment with $\epsilon = 10^{-3}$ can be found in Figure 7 in Appendix B.

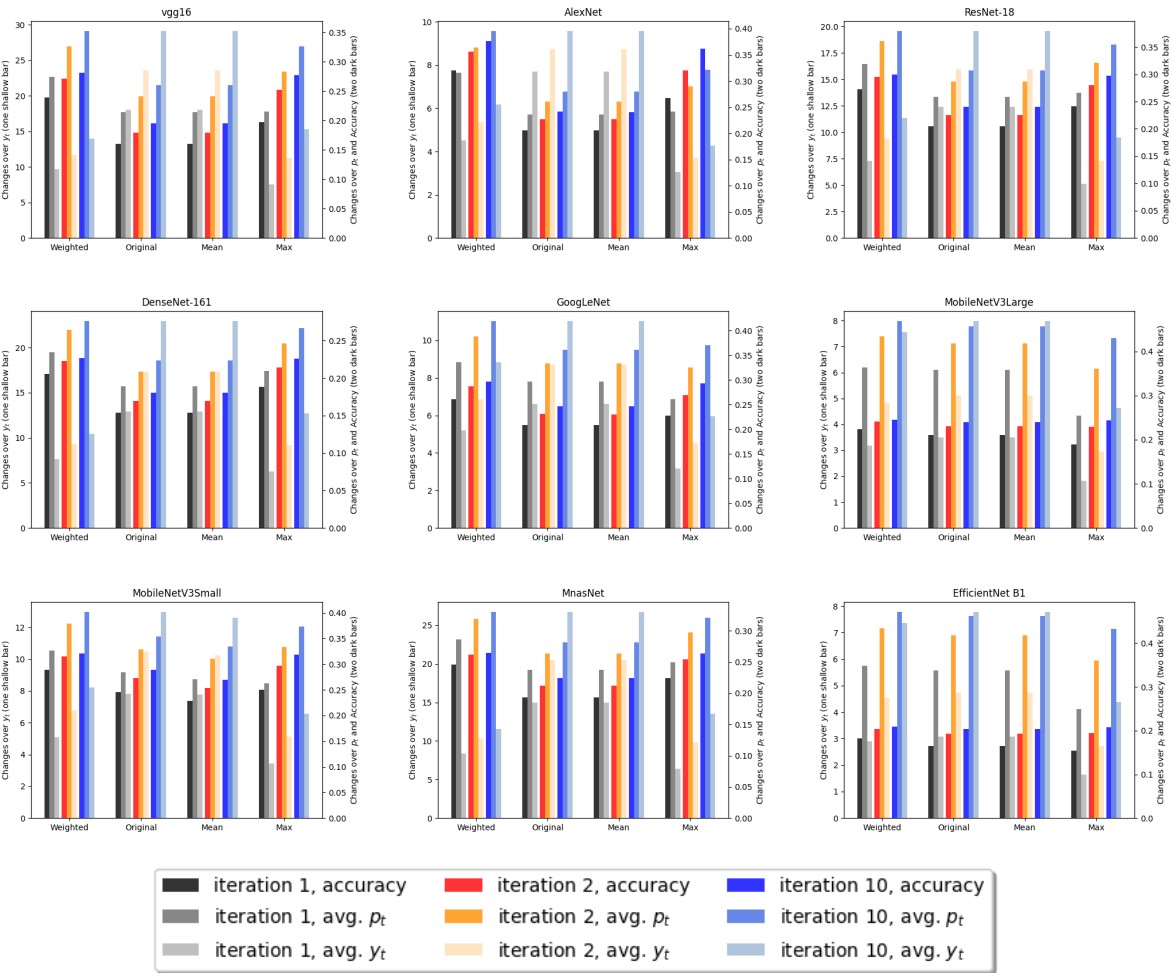

Figure 1: Reproducing of Figure 3 in the original paper with $\epsilon = 3 \times 10^{-3}$. Changes in accuracy, $y_t$ and $p_t$ (t is the target classification class) when certain input features are perturbed. Perturbed features are selected based on four gradient explanations (original, mean, max and weighted), where original is directly with respect to the gradients of the logits.

Furthermore, the equations for the gradient sign perturbation in the original paper turned out to have errors in the clamping and indexing of the iterations. The correct equations are

$$\boldsymbol{x}^{n+1} \leftarrow \boldsymbol{x}^n + \alpha \operatorname{sign}(\phi^t(\boldsymbol{x}^n)) \tag{5}$$

$$\boldsymbol{x}^{n+1} \leftarrow \operatorname{clamp}(\boldsymbol{x}^{n+1}, \max(\boldsymbol{x}^0 - \epsilon, 0), \min(\boldsymbol{x}^0 + \epsilon, 1)) \tag{6}$$

where $n$ is the number of iterations, $\epsilon$ is the perturbation limit, and $\alpha = \frac{\epsilon}{n_{tot}}$ is the step size, $n_{tot}$ is the total number of iterations.

Our results verify the results reported in the original paper and are evidence for Claim 1, since the weighted and max explanation methods yield an increase to $p_t$ and accuracy, while the original and mean explanation methods yield an increase to $y_t$. Although the results are similar to those of the original paper there are some numerical differences in Figure 1 which is probably due to different weights in the models and hence also different original performance.

### 3.2 Reproducing 5.2 Back-Propagation till the Activation Space

This section reproduces section 5.2 in the original paper by performing the same experiments of both visualization and effects of blurring and masking. These experiments were all performed on VGG-16 with batch normalization (Simonyan & Zisserman, 2014) fine-tuned on the CUB-200 dataset (Wah et al., 2011). The fine-tuning was done with an SGD optimizer with momentum using a batch size of 128, learning rate of $10^{-3}$, momentum of 0.9, and weight decay of $5 \times 10^{-4}$. The model was trained for 200 epochs on the training set as defined by the dataset. For an exact implementation or to reproduce the model, see our repository. The results of this section generally show evidence for Claim 2, both qualitatively and quantitatively, and show that the proposed weighted contrastive method highlights crucial areas for classification when the model classifies between several dominant classes. The extensions to XGradCAM and FullGrad also show generalizability of the method and thus strengthens Claim 3.

#### 3.2.1 Visualizations

Reproduction of the visualizations of three different back-propagation-based methods can be seen in Figure 2. Here we compare GradCAM and Linear Approximation, as described in the original paper, and XGradCAM, as described in section 2.1, to their contrastive weighted counterpart, which was obtained by back-propagating from the softmax neuron $p_t$ of the target class $t$ rather than its logit $y_t$. The visualization was done by overlapping the bilinearly interpolated relevance map on top of the original image with an alpha of 0.5. A centered norm was applied on the heatmap before visualizing using the `bwr` colormap in `Matplotlib`. The images were picked such that $p_2 > 0.1$ and were selected at random to prevent bias from only selecting good samples. Observe that the samples picked are different from those in the original paper as those samples did not have a probability for the second most probable class over the threshold.

The results are partly in line with what the original paper suggests. Firstly, one can note that the original explanation method is quite consistent among the two classes with differences being mostly the intensity of the positive and negative areas. Secondly, one can also see that the weighted methods produce almost complementary heatmaps for the two classes, which makes sense as they are mostly dominant over all other classes. Lastly, we see a large difference in the size of the negative and positive areas visualized compared to the original paper. This is presumably due to different methods of visualization, but as the procedure of visualization of the original paper was not detailed this cannot be confirmed. Observe that the large negative areas in some images, especially seen when comparing our GradCAM to other implementations, are due to the omission of ReLU as described in the original paper. Our results therefore also conflict with the claim in the original paper in appendix G, where the authors claim that non-contrastive methods have much larger positive than negative areas. In Figure 2 one can see that the original GradCAM has much larger negative areas than positive for all selected images.

The same experiments when performed using FullGrad produce fully negative saliency maps. The modified FullGrad is therefore not truly contrastive as it does not have both positive and negative contributions instead one has to use normalization and assume that they are evenly distributed. When normalizing is applied to the final saliency map the results are similar to those seen in Figure 2 and some select images can be seen in Figure 3. These seem to be of a more fine-grained nature than the GradCAM-based methods in Figure 2 while largely highlighting the same areas. This suggests a suitable alternative to GradCAM-based methods and that a contrastive visualization is possible for FullGrad but that this relies on normalization.

#### 3.2.2 Blurring and masking

Reproduction of the blurring and masking experiment seen in Table 1 of the original paper can be seen in Table 1. Here we also added an additional row with results using XGradCAM. FullGrad is not analyzed as the modified version only produces negative areas. This gave similar results to GradCAM and Linear Approximation although performed slightly better on the negative features and for positive features for the second most probable class $t_2$. Here we use the same baselines as the original paper with the motivation of them having slightly different results without a generally accepted standard (Sturmfels et al., 2020). The values in the table are the average relative probability of the most and second most probable classes for each image. This relative probability is defined as $\bar{p}_{t_i} = \mathbb{E}\left[ e^{y_{t_i}} / (e^{y_{t_1}} + e^{y_{t_2}}) \right], i = 1, 2$ where $t_i \in [c]$ represents

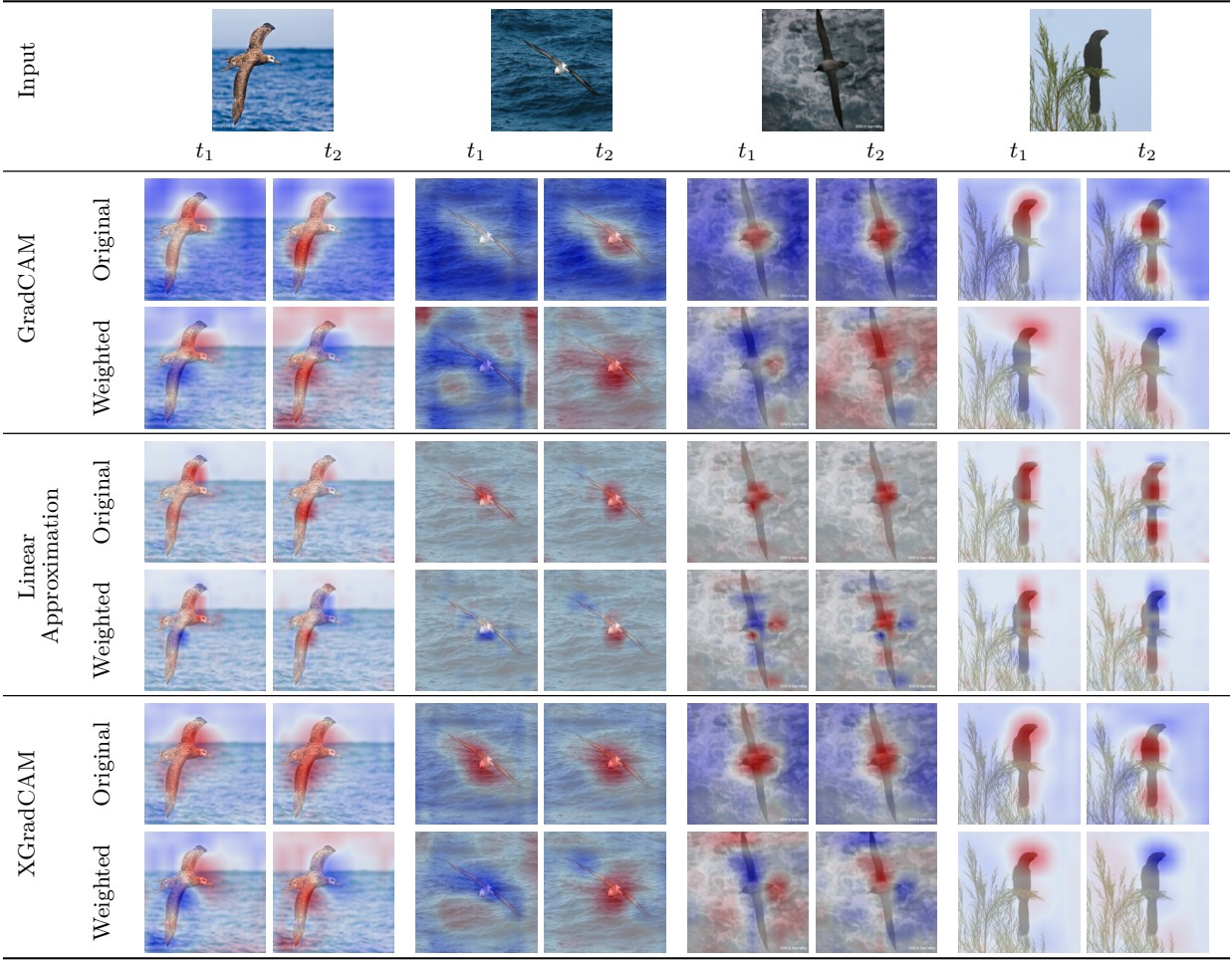

Figure 2: Reproduction of Figure 4 in the original paper. Comparison between the back-propagation from logits $y_t$ (Original) and weighted contrastive back-propagation from $p_t$ (Weighted) for GradCAM, Linear Approximation, and XGradCAM. The columns for each image signify the most possible and second possible class, respectively. Red and blue signal positive and negative activations respectively.

the $i$-th most possible class. These expectations are, like in the original paper, only calculated over samples that fulfill the threshold criteria $p_2 > 0.1$.

The results are very similar to those of the original paper, although not identical, and show the same patterns. We decided to use equal blurring and masking here to prevent bias where one method might yield larger or smaller negative areas to guarantee that the original and weighted methods both modify an equal number of pixels. This was also suggested in the original paper in appendix G and seems to have a minor impact on the results while negating some bias.

### 3.3 Reproducing 5.3 Comparison with Mean/Max Contrast

We perform the same experiments as in section 5.3 of the original article. Here we reuse the same VGG-16 model used in section 3.2 and implement mean and max contrast as described in the original paper. The used method for visualization is also the same as in section 3.2 and a threshold of $p_3 > 0.1$ is used. The results, seen in Figure 4, are similar to the original paper, especially the observation that original and mean methods yield extremely similar results due to the tiny scaling factor used when subtracting by the other classes in the mean method. We also note that max similarity for the two most probable classes is each other's inverse and that the weighted method gives a similar but more detailed comparison that includes

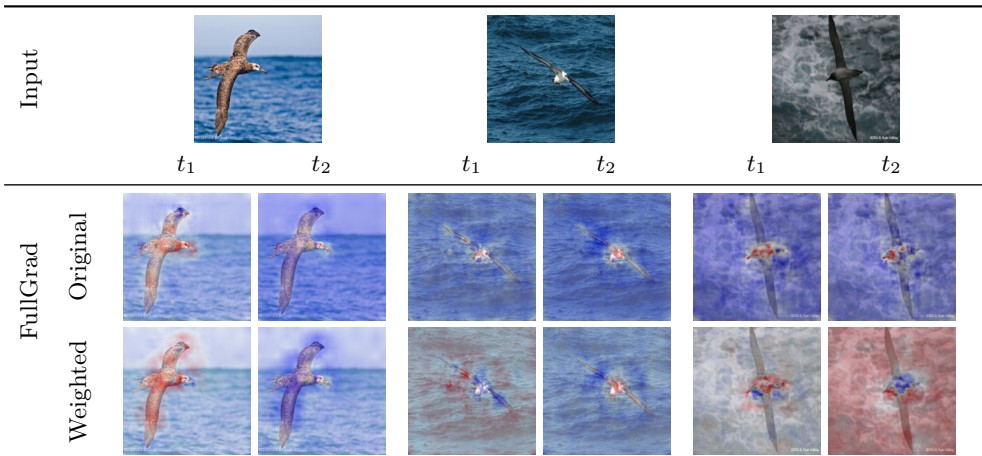

Figure 3: Comparison between the back-propagation from logits $y_t$ (Original) and weighted contrastive back-propagation from $p_t$ (Weighted) for FullGrad. The columns for each image signify the most possible and second possible class, respectively. Red and blue signal positive and negative activations respectively after normalization.

Table 1: Reproduction of Table 1 in the original paper using equal blurring. Comparisons between weighted contrastive method (wtd.) and original method (ori.) when blurring and masking. Using baselines Gaussian Blur, Zeros, and Channel-wise Mean and the methods Linear Approximation (LA), GradCAM (GC), and XGradCAM (XC). $t_1$ and $t_2$ are the classes with the highest and second highest probability respectively. Each line shows how the average relative probability changes among each image's top two classes. Pos. and Neg. Features mean that only positive and negative features are kept with respect to the corresponding target class. It is expected that when the positive or negative features corresponding to the target are kept, the expected relative probability is expected to increase or decrease respectively.

|  |  |  | $p_t$ | Gaussian Blur | | | | Zeros | | | | Channel-wise Mean | | | |
| --- | --- | --- | --- | --- | --- | --- | --- | --- | --- | --- | --- | --- | --- | --- | --- |
|  |  |  |  | Pos. Features | | Neg. Features | | Pos. Features | | Neg. Features | | Pos. Features | | Neg. Features | |
|  |  |  |  | ori. | wtd. | ori. | wtd. | ori. | wtd. | ori. | wtd. | ori. | wtd. | ori. | wtd. |
| CUB-200 | LA | $t_1$ | 0.712 | 0.695 | **0.789** | 0.419 | **0.274** | 0.663 | **0.754** | 0.428 | **0.292** | 0.676 | **0.766** | 0.426 | **0.281** |
|  |  | $t_2$ | 0.288 | 0.560 | **0.738** | 0.390 | **0.211** | 0.563 | **0.717** | 0.398 | **0.253** | 0.558 | **0.729** | 0.391 | **0.235** |
|  | GC | $t_1$ | 0.712 | 0.747 | **0.858** | 0.428 | **0.271** | 0.731 | **0.850** | 0.432 | **0.286** | 0.745 | **0.857** | 0.426 | **0.277** |
|  |  | $t_2$ | 0.288 | 0.461 | **0.759** | 0.402 | **0.199** | 0.469 | **0.761** | 0.414 | **0.226** | 0.468 | **0.759** | 0.406 | **0.214** |
|  | XC | $t_1$ | 0.712 | 0.733 | **0.847** | 0.422 | **0.248** | 0.711 | **0.838** | 0.426 | **0.266** | 0.719 | **0.844** | 0.419 | **0.253** |
|  |  | $t_2$ | 0.288 | 0.504 | **0.785** | 0.393 | **0.169** | 0.515 | **0.777** | 0.402 | **0.184** | 0.511 | **0.784** | 0.395 | **0.177** |

several classes simultaneously. Like in section 3.2 we also observe that the negative areas are much larger than in the compared article, presumably due to different visualization methods.

Figure 4 also highlights the strengths of the weighted contrastive method. Here it is clear that the weighted method helps give detail to which areas of the image are key for a specific classification given a choice of several dominating classes. This can be useful when debugging misclassified samples where positive regions using the weighted method indicate regions that the model considered in its choice. For example, for the top-left part of Figure 4 one can clearly see that the top class puts a heavy bias on a few select spots of the background, thus indicating that the model might be utilizing non-object pixels to classify the object. This is further evidence for Claim 2.

## 3.4 Vision Transformers and contrastive GradCAM

To adapt GradCAM to Vision Transformer models the outputs of the multi-head attention blocks of the ViT are assumed to be spatially coherent nodes as in standard CNN models. This is convenient as they generally have the same dimensionality as the input patches, here 16x16. This means that instead of backpropagating toward a convolutional layer GradCAM backpropagates toward a late multi-head attention block. This results in a 16x16 explanation map after taking the mean of the channels, where channels here are not RGB

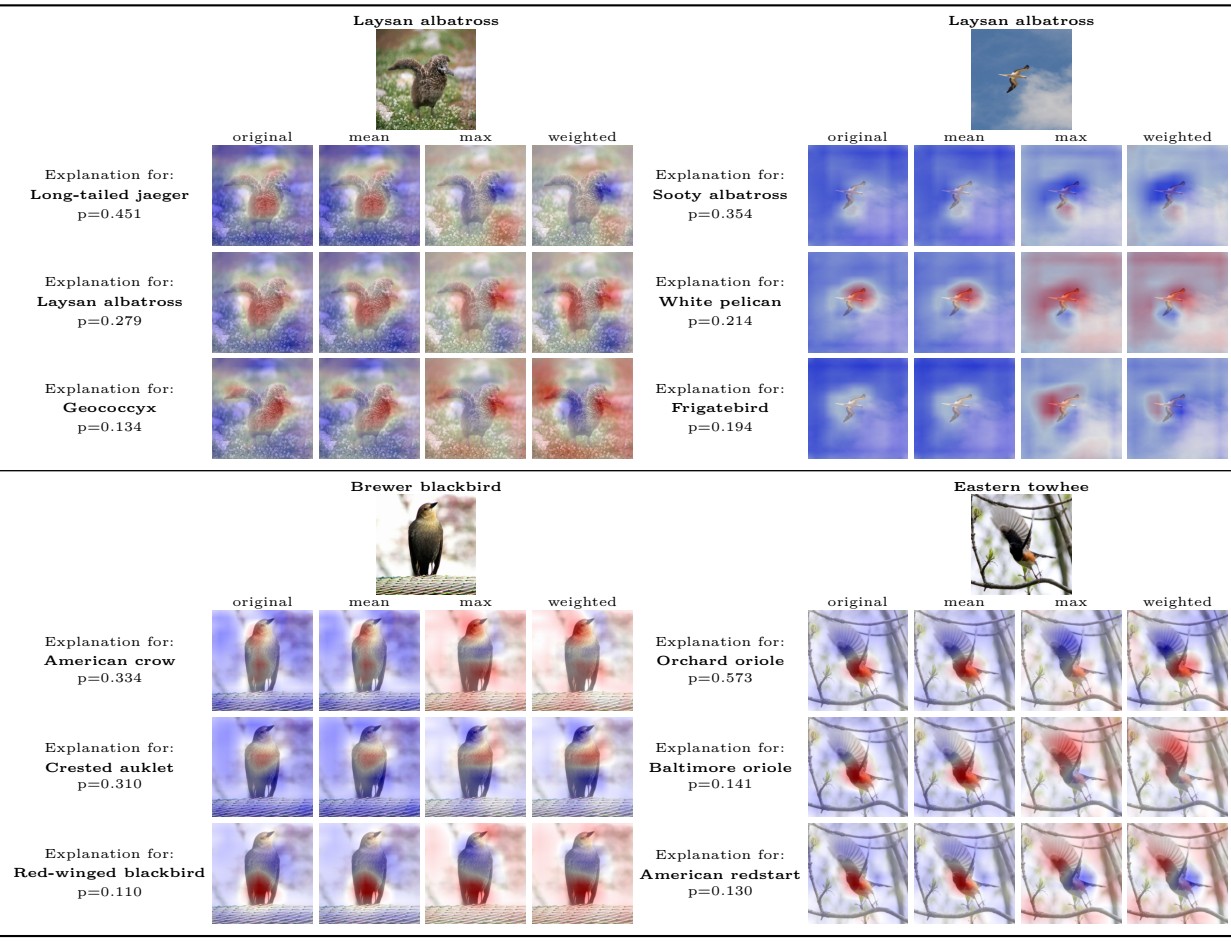

Figure 4: Reproduction of Figure 5 in the original paper. Comparison between mean, max, and weighted contrast for four images from CUB-200. In each column, we present explanations for the three most probable classes for GradCAM using the original image and the three contrastive methods.

channels as in CNN but the embedded dimension of the tokens. These explanations are then upsampled to the original image's size. For a more detailed description of how this is implemented, see Gildenblat & contributors (2021).

ViT models process information from pixels differently from CNNs. While CNNs inherently have a spatial connection between input pixels and activations, enforced by limited filter sizes, [1] this spatial relation is not enforced in ViTs. The self-attention module in ViT allows them to attend to and be influenced by patches, or tokens, regardless of distance. It has been shown that contrary to CNNs, ViT models attend to pixels regardless of distance from the first layer Dosovitskiy et al. (2020). For evaluating this we use the model implemented in `PyTorch`[2] and fine-tune it on the Food-101 dataset (Bossard et al., 2014). Initial attempts were also made without fine-tuning evaluating on ImageNet, as can be seen in Appendix B, although these results are less clear as the dataset is not as fine-grained.

We get qualitatively worse results compared to CNNs, with most explanations generating nonsense results that do not seem to be correlated to the image. We believe that this is mostly due to the weaker spatial relationship between token-wise representations and that the method for upscaling patches, or activations, in later layers, to input image does not adequately represent pixel importance in ViTs. The alternative method

---

[1]Filter sizes in CNNs are usually not larger than $7 \times 7$, therefore the spatial distance between the two pixels influencing an activation can at most be 7.

[2]See, using the default weights, https://pytorch.org/vision/main/models/generated/torchvision.models.vit_b_16.html.

of Gradient-weighted Attention Rollout is considered in Section 3.5 as a partial solution to the spatial mixing problem.

A few examples of good explanation maps can be found in Figure 5a but these are rare and selected from the multi-head attention blocks that for those images gave spatially coherent results which can vary between images. We find that the contrastive explanation does affect the results, giving more detail in the highlights as can be seen in the pad thai and rice example in Figure 5a.

### 3.5 Vision Transformer: Contrastive Gradient-weighted Attention Rollout

To alleviate the problem of hard-to-find proper explanations due to less enforced spatial coherence, explanations through attention rollout are attempted. Attention rollout as an explanation method for ViT was proposed in Dosovitskiy et al. (2020), with the theory laid out in Abnar & Zuidema (2020). With attention rollout, information flow is quantified by backtracking the attention from the desired layer to the input layer by multiplying the attention matrices. This partially restores the information flow between tokens and patches in ViT. This method has later been further developed in order to weight explanations with regard to their gradients (Gildenblat, 2020; Chefer et al., 2020), similar to GradCAM.

The gradient-weighted attention rollout explanation is constructed from the gradient-weighted attentions of each layer, defined as the mean over the attention heads of the gradient with regard to the target logit elementwise multiplied with the attention activation. These gradient-weighted attentions are then propagated down towards the input by multiplying these matricies together.[3]

This explanation is significantly more accurate to the perceived localization of the image. For example, one can clearly see in Figure 5b that the method highlights rice and noodles for the different classes respectively. The weighted contrastive method with regard to the softmax further shows an even more detailed explanation. This is especially obvious when the dominating classes are of similar probability as in the pad thai and rice example shown in Figure 5b. In other cases, such as in the sushi and ramen example, where there is one dominating class but many probable classes with $p_t \approx 0.05$ the weighted contrastive version is similar to the normal version. Overall this shows that a ViT implementation of the proposed contrastive weighted method is possible and relatively easy to implement, thus strengthening generalizability and Claim 3.

## 4 Scientific considerations

### 4.1 Reproducibility

As seen in the checklist of the original paper no large efforts were made toward the reproducibility of the paper. For example, no links to working code or details on the fine-tuned model training were provided. This heavily impacted our work as we had to make many assumptions about the process. We did find, however, a repository at https://github.com/yipei-wang/ClassContrastiveExplanations/ that contained some code regarding the fine-tuning of VGG-16 on CUB-200. This helped in specifying hyperparameters that would reflect those of the original paper. This also showed that they used VGG-16 with batch normalization, which was not specified in the original paper and the difference compared to the non-batch normalized variant will yield different results.

Lack of code or detailed method also led to difficulties reproducing some results, as seen in section 3 especially coupled with some errors. For example, the inaccurate equation in section 5.1 in the original paper coupled with the wrong epsilon led to many difficulties in reproducing and understanding that section. It is also not specified as to which data the fine-tuned models are trained. There are also some minor mistakes such as the bird in Figure 1 of the original paper having the wrong input label.

We also found it unclear during our first readthroughs of the article that the authors' weighted contrastive method should ideally be implemented by back-propagating from the $p$ neuron and the performance gains that this gives. In general, the presentation of their weighted contrastive method as novel led us to miss the

---

[3]Gradient-weighted attention rollout has been implemented in `https://github.com/jacobgil/vit-explain/blob/main/vit_grad_rollout.py`

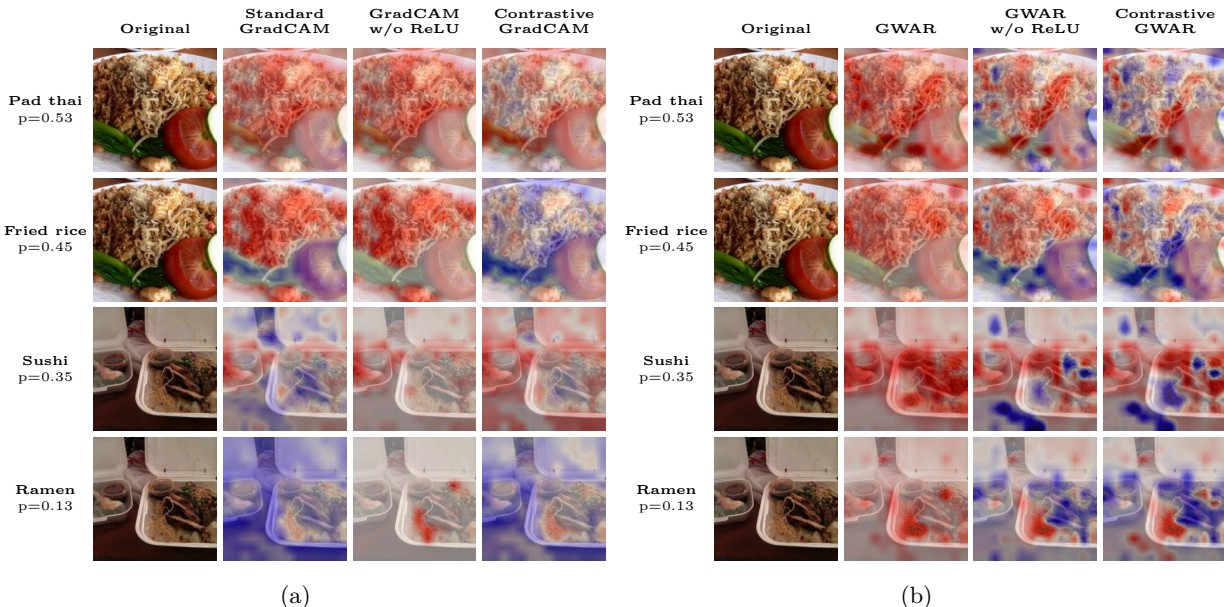

(a)                                                                  (b)

Figure 5: Comparison between proposed explanations. In (a) a comparison between GradCAM, GradCAM without ReLU, and Contrastive GradCAM is considered with target attention layer 8 and 10 respectively. In (b) a comparison between Gradient-weighted Attention rollout (GWAR) of the standard, without ReLU, and contrastive variant is considered. Red sections are considered areas with high explainability. To adapt the method to the contrastive version all ReLU operations were removed and the gradients were calculated from the softmax output instead of the logits.

conclusion that it was proportional to back-propagating from the $p$ neuron for many explanation methods. Our experiments show, however, that for more sophisticated explanation methods more adjustments have to be made to the original method in order to make it contrastive by introducing negative areas.

## 4.2  Scientific contribution

The original paper provides an intuitive and efficient way of generating contrastive explanations that can take multiple classes into account. They also show that these outperform generally non-contrastive methods regarding the strength of the probability for the target class. They do not, however, make any large comparisons to state-of-the-art baselines in contrastive explanations. They defend this in peer reviews by claiming that many other contrastive methods ask the question of "how to change the instance to other classes?" while the authors aim to answer "why the instance is classified into this class, but not others?". Furthermore, many other contrastive methods are only suitable for preliminary data such as MNIST rather than the high-resolution images used here. Therefore we deem this lack of comparisons to other methods as valid.

Another observation is that all results rely on the class probability $p$ as a metric for the relevance of the explainability method. While this is intuitive it also seems obvious that the contrastive weighted method presented which back-propagates from the $p_t$ neuron will outperform the same method based on the preceding $y_t$ neuron. This makes the results very expected, especially the ones shown in Figure 1 and Table 1. The visualizations show, however, that this method yields a clear explanation as to which areas of the image are especially important for a certain class, and in the end, this is perhaps the greatest contribution.

We also find that the authors' work is more of an explanatory nature than inventing something novel, as back-propagating from the $p$ neuron has been commonly done before and even mentioned in the original GradCAM paper (Selvaraju et al., 2019). The value is therefore showing that back-propagating from target

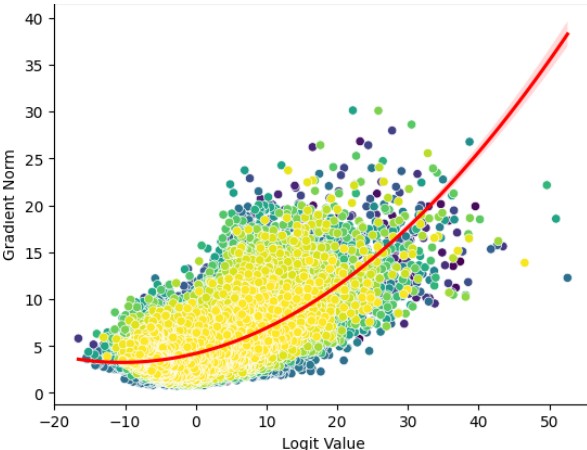

Figure 6: Relationship between the explanation weight $\|\phi_y\|$ and logit value $y$, using $\phi_y = \nabla_x y$ as an explanation, for 145 target classes for 1000 random images from ImageNet. A second-degree regression is applied and shows a clear upward trend (red). Colors are applied solely for visibility.

softmax neuron $p_t$ yields a proven weighted contrastive explanation compared to back-propagating from logit $y_t$.

### 4.3 Dominating classes

The authors have explicitly chosen not to do experiments on images where there exist dominating classes where $p_1 \gg p_2$. This is not motivated in the paper but is likely because the contrastive weighted method tends to be reduced to the original, non-contrastive, explanation under such circumstances. This is easy to make note of during testing when looking at images where $p_2 \approx 0$ where the contrastive and non-contrastive versions show no qualitative difference. Table 2 also shows a reproduction of Table 1 but inverting the threshold and only using samples where the second most likely class has a probability $p_2 < 0.1$. This table clearly shows that the positive features between the original and weighted method are on average very similar while the negative regions in the weighted variant are slightly more effective in decreasing the target probability, although less so than in Table 1.

That the explanation methods are so similar for low $p_2$ can be explained by $p_2$ often being very low, $< 0.001$, and much closer to $p_3$ than $p_1$. In those cases the weighted method when targetting the most likely class $t_1$ the subtracted weighted sum in Equation 1 will go toward zero as the non-target classes take out each other. As seen in Table 2 this seems to mostly apply to the positive features of the weighted method and therefore it seems that the negative features of the non-target classes seem to be taking out each other.

Another reason for this behavior is explained by the observed relationship that explanation weights seem to increase with logit strength or output probability. This is exemplified in Figure 6. Due to this, we can expect an explanation with regards to the dominating value to be weighted significantly higher, around 3 to 4 times, than all other features. For future work, normalizing the explanation before weighing could be considered.

The behaviour of $\frac{\partial p_i}{\partial y_j}$ should also be considered. Here we observe that if the softmax output is dominated by a single value the gradient goes to zero. That if there exists a $p_t \rightarrow 1$ then it implicates $\frac{\partial p_i}{\partial y_j} \rightarrow 0$, this is easily observed in the Jacobian.

One could also argue that there is no use of contrastive methods when there is only a single dominating class as then the model is certain in its decision and is not weigh the possibility of another class. However, in scenarios where a misclassification has occurred a contrastive method to compare the correct class to the misclassified class can be useful, thus there is a use for contrastive method even without dominating classes showing an opportunity for advancement in future work.

Table 2: Reproduction of Table 1 using threshold criteria $p_2 < 0.1$ for GradCAM (GC). The probability of the second most likely class is almost always $p_2 \approx 0$ with an average $\bar{p}_2 = 0.01$, and has been omitted for this reason as it is mostly ambiguous.

| | | | $p_t$ | Gaussian Blur | | | | Zeros | | | | Channel-wise Mean | | | |
| | | | | Pos. Features | | Neg. Features | | Pos. Features | | Neg. Features | | Pos. Features | | Neg. Features | |
| | | | | ori. | wtd. | ori. | wtd. | ori. | wtd. | ori. | wtd. | ori. | wtd. | ori. | wtd. |
|---|---|---|---|---|---|---|---|---|---|---|---|---|---|---|---|
| CUB-200 | GC | $t_1$ | 0.985 | **0.981** | 0.979 | 0.442 | **0.351** | 0.973 | **0.974** | 0.438 | **0.358** | **0.977** | **0.977** | 0.437 | **0.349** |

## 5  Conclusion

Overall the paper provides a clear argument as to how back-propagating from the softmax prediction instead of the logits gives improved connectivity to the actual prediction and thus a more relevant contrastive explanation. They propose a simple way of implementing this, which is applicable to many models and methods, and it shows a clear connection to accuracy using removal and blurring metrics. Their method also answers why a sample is predicted to belong to a certain class above others.

We have reimplemented their work, made some corrections to their method, and have further been able to apply their method to other similar tasks using Vision Transformer architectures and the XGradCAM and FullGrad explanation methods with good results. Due to the simple nature of their contrastive method, one can also easily reproduce it by using back-propagating explanation methods from after the softmax layer which makes it generally reproducible. Some methods might, however, require simple modifications such as removing ReLUs or somehow introducing contrastiveness such as through normalization.

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
