# OpenReview forum: "Reproducibility review of “Why Not Other Classes?”: Towards Class-Contrastive Back-Propagation Explanations"
_TMLR — Rejected by TMLR_

### Review · Reviewer_Nhj9 · 2024-03-09

**Summary Of Contributions:**

This paper focuses on replicating the experiments presented in " 'Why not Other Classes?': Towards Class-Contrastive Back-Propagation Explanations". The authors re-implement the experiments from original study, and also introduce some corrections and refinements to the methodologies and formulations. The study verifies the experimental results reported in the original paper and it additionally broadens the scope by extending the implementation to Vision Transformers (ViTs). As a reproducibility study, this paper serves as an effective complement and extension to the original work.

**Audience:**

Yes

**Broader Impact Concerns:**

This work does not contain any broader impact concern.

**Claims And Evidence:**

Yes

**Requested Changes:**

Please refer to the **weakness** above.

**Strengths And Weaknesses:**

**Strengths**
1. This paper systematically evaluates the targeted work in many aspects, including the experiments, methodologies and formulations.
2. The authors also extend the original approach to other methods such as XGrad-CAM and other model families such as ViTs. Additionally, their discussions on the results can also be a complement to the original paper.
3. The presentation is clear and easy to follow.
4. The code of all the implementations is also made available.

**Weaknesses**

Overall, this work has no significant technical weaknesses. As a reproducibility study, the results are convincing and comprehensive. The discussion is also easy to follow. However, some minor details can be further improved as follows:

1. The extension to ViTs is described too vaguely. It is suggested that the authors add more details to enhance readability. For example, in the first paragraph of section 3.4, "tokens are reimagined as the spatially ... these explanations are later upsampled to the original image’s size." Some illustrations or formulations can improve the readability here.
2. The arrangement of the figures appears arbitrary. For example, in page 8, the arrangement and spacing of Figure 4 is as informal as an appendix. Figure 6 also seems to be truncated.
3. In Figure 6, what does the colormap of the points signify? And how is the red curve obtained?
4. Please revise the citation format, e.g. "(Gildenblat, 2020; Chefer et al., 2020)" in section 3.5; \citep instead of \citet in section 3.1.

---

> ### Author Response · Authors · 2024-03-15
>
> Thank you for your review.
>
> We agree that the descriptions in section 3.4 and 3.5 were described very vaguely and have made a revision where it is rewritten. While illustrations would probably be more educational, we instead refer to a source where such illustrations are present as it feels slightly out of scope for a reproducibility review which is already quite long.
>
> Thank you for also pointing out some mistakes with our figures, this is hopefully also remedied in the revision.
>
> Best regards,
> Authors

---

### Review · Reviewer_tXr9 · 2024-03-13

**Summary Of Contributions:**

The paper is a reproducibility review of "Towards Class-Contrastive Backpropagation Explanations" (Wang & Wang 2022). It validates the main claims of the original paper - mainly that weighted contrastive explanations are equivalent to backpropagating from the softmax rather than the logits, conducts a thorough review of the experiments of the original paper, and further extended them by:
* Testing the approach with a different explanation methods,
* Testing the approach with a different family of architectures.

The authors conclude that pending some minor errors, the method is simple and easy to reimplement, and trends in the original paper are generally reproducible. There were numerical differences with the original paper that are explained by potentially different model weight, the lack of code and potentially different hyperparameters due to the lack of some experimental details in the original paper.

**Audience:**

Yes

**Broader Impact Concerns:**

The work is a reproducibility review. Discussing the broader impacts of the original paper could make the paper stronger, but I don't see it as a requirement for this work.

**Claims And Evidence:**

No

**Requested Changes:**

Based on the Weaknesses listed above, I suggest the following:
* Test the approach with another explanation method, more significantly different from GradCAM.
* Provide a more detailed discussion of the experiments with the visual transformers.
* Review section 4 (details above). I think the claims in these section need to be made more precise or supported better.
* Reconsider the evaluation approach.

**Strengths And Weaknesses:**

Strengths:
* The reproducibility of the original paper is reviewed in a systematic and thorough manner. The scope is clearly stated. The authors furthermore seem to have sought details from the authors of the original paper for details on which they have doubts (e.g. the value of maximum perturbation in the gradient sign perturbation (eq (5) and (6)).
* The paper not only tackles the reproducibility of the original paper, but also its generalizability. I really appreciate this effort.
* The paper noted an error in the gradient sign perturbation equation in the original paper (sec. 5.1). They corrected it and made it more precise.
* The authors provide a clear discussion of the original paper contributions in section 4.

Weaknesses:
* I think while the intention of studying generalizability of the approach is great, both angles are not pushed far enough. The other studied explanation method is a modification of GradCAM that was already thoroughly studied in the original paper. It would be interesting to analyse another explanation method, such as FullGrad (Srinivas & Fleuret 2019) or another attribution based method. The experiment with the visual transformers are interesting, but could be analysed a bot more. The authors observed that for these models, the contrastive explanations give similar results to their original versions. What could explain this? Is it an effect of the attention mechanisms, or of the way explanation methods have been implemented for these models to enforce spatial coherence?
* In the scientific considerations section, the authors present the main contribution of the paper as producing explanations by backpropagating from the probabilities rather than the logits. This seems unfair to me. The authors present a general principle that turned out to be equivalent to backpropagating from after the softmax in many cases, which made it also practical. This doesn't mean that this is always the case, and the approach is general enough to be generalisable to other cases. The general approach is a motivation to consider more often backpropagating after the softmax..
* The authors noted that the original paper doesn't contain experiments where there is only one dominating class. They claim that this is likely because the advantages of the approach are reduced. This claim is not supported by experimental results. It also seems unfair, as a more plausible explanation is that it is normal for these cases to not benefit from this contrastive approach. The question of "why not other classes" is of lower relevance. It is however less obvious that the contrastive approach would hurt.
* The authors also picked up another weakness of the original paper, without rectifying it. The evaluation of the explanation methods in both papers relies mostly on visualisations and non-robust metrics. "Towards Better Visual Explanations for Deep Image
Classifiers" (Grabska-Barwinska et al. 2021) offers a discussion of the evaluation of saliency maps [Section 4].

---

> ### Author Response · Authors · 2024-03-26
>
> Thank you for your valuable feedback.
>
> To answer the stated weaknesses:
>
> Q1) We agree that both angles were not pushed far enough and that the results for ViT were underwhelming and saw almost no difference between the original and the contrastive method. To remedy this we have now performed evaluation using FullGrad (which we as of doing the experiments were unaware of, thank you for bringing this to our attention) to some success, although an issue here is the conversion of the post-processing operations in FullGrad which in the paper depends on the non-linear abs() function to operations that are linear and therefore compatible with the proposed weighted contrastive. We remedy this by simply removing the abs() function, although we note that during aggregation of the different gradients the resulting saliency map will always be negative, and thus we apply normalizing (which also naturally loses some information of which attributions are truly positive/negative).
>
> Regarding ViT, after some contemplation, we realized that our results did not make much sense and after some debugging and further experimentation with a fine-grained dataset (Food-101) we found good results where contrastive properties can clearly be seen when compared to the original methods. We do not believe that the attention mechanisms themselves affect the difference between the original methods and the weighted contrastive versions, but that this is mostly attributed to many low probability but still probable classes (i.e. see Sushi/Ramen example for GWAR in Figure 5 in the updated version).
>
> Q2) We agree that the author's main contribution is the contrastive weighted method itself and that they show that it often, but not always, can be achieved by efficiently backpropagating from after the softmax layer. Our presentation was a bit odd in the paper and we have remedied this in the new revision and further emphasized the weight of their contribution by showing that it can quite easily be applied to new domains.
>
> Q3) We have now made further experiments strengthening the reduced impact of the weighted contrastive method when there is a single dominating class (see Table 2 in the updated version). From that, we saw that the positive areas of the contrastive weighted method are basically reduced to the original method - while the negative areas still carry some new information. We also do think the question of "why not other classes" can still be of relevance with a single dominant class if this class is a misclassification. In that case, one might be interested in comparing the classified class to the true class and seeing the differences that made it prioritize the true class. We added a discussion regarding this to the revised version.
>
> Q4) We acknowledge the critique regarding our method's reliance on visualizations and non-robust metrics for evaluating explanation methods. While we understand the significance of employing more rigorous evaluation metrics, we have not been able to find or construct a metric that does not suffer from the same issues regarding a basis on softmax accuracy, as we argue that these metrics implicitly favor the gradient-based explanations that are based on the same softmax output. This also applies to (Grabska-Barwinska et al. 2021). One can also discuss whether or not this is actually an issue, as after all what we are interested in is often really the classification confidence, which these metrics capture.
>
> Please let us know if there is anything else you are wondering about regarding our work!
>
> Best regards,
> Authors

---

### Review · Reviewer_h5Nh · 2024-03-13

**Summary Of Contributions:**

This paper studies the reproducibility of a previous paper (Wang & Wang 22).
It finds that Wang & Wang 22 lacks details for reproducibility, and has an error in an equation about input perturbation.

The paper also studies the generalizability of Wang & Wang 22 by extending results to other setups:
- a different interpretability method, i.e. XGradCAM, for which the weighted contrast method provides reasonable generalization.
- a different architecture, i.e. Vision Transformer, on which it was found that the weighted contrast method cannot provide meaningful feature map, likely due to the mixing in attention.

**Audience:**

Yes

**Broader Impact Concerns:**

There are no direct ethical concerns.

**Claims And Evidence:**

No

**Requested Changes:**

- Please update the paper to have much more contexts and motivation. Please also highlight your findings, e.g. summarizing which findings in Wang & Wang 22 were confirmed/rebutted by the current paper.
- To improve the contribution and impact, please provide more new findings and insights, possibly by looking beyond what was considered in Wang & Wang 22.

**Strengths And Weaknesses:**

On the positive side, the current paper closely examines several setups in Wang & Wang 22, and provides an open-source implementation of the weighted contrast method proposed in Wang & Wang 22, which is of value to the community.

For suggestions, I want to first comment on the writing/organization: The paper should be made much less about problems in Wang & Wang 22 and much more about new findings and insights.
- Perhaps due to the fact that this paper is focused on Wang & Wang 22, the current paper lacks contexts that would make the paper appear insufficiently motivated and also likely confuse readers who have not seen the previous paper.
  - For example, the intro could include more motivation. intuition and details about the weighted contrast method.
  - Please provide background on GradCAM and Linear Approximation, perhaps in the appendix if the space is a concern.
  - Notations such as $y$ (logit) and $p$ (probability) could be more clearly defined.
  - In Section 3, the paper refers to sections in the previous paper but does not provide sufficient details what the methods are (e.g. "four different explantation methods" were not explained).
- To better motivate the current study, the intro should clearly state the shortcomings of the weighted contrast paper. It would also be helpful to summarize the main findings in the current paper. Similarly, to better contrast the current paper to the previous one, subsections in Sec 3 should clearly state the main takeaways, such as the main limitations (of the previous paper) to address and the main findings of the current paper.
- Another way to highlight key findings is to make the (sub)section titles more informative; e.g. rather than "reproducing Section X", say "reproducing {something}: {takeaway}".


It would also be interesting to study setups outside of what was considered in Wang & Wang 22.
- For example, in Sec 3.2.1, what would happen when the second class probability $p_2$ is lower (i.e. as Sec 4.3 also points out)? Are samples with incorrect predictions considered? On the incorrectly predicted samples, is weight contrast helpful in understanding / debugging the model?
- The explanation maps are found to be of poor quality for ViTs, possibly due to the mixing in self-attention. Do you have suggestions/insights on how to fix this, or proposal on what interpretability methods work for ViT (e.g. perhaps along the line of attention rollout)?

---

> ### Author Response · Authors · 2024-03-26
>
> Thank you for your review!
>
> To answer your suggestions:
>
> Q1) We agree that our original submission lacked context which made it depend heavily on the original paper. We have remedied this by including more motivation in the introduction, adding some background for GradCAM and Linear Approximation (in the appendix as suggested), clarifying notations regarding y and p when they are used, and providing more detail in Section 3 as mentioned.
>
> Q2) We have updated the introduction to describe the original paper in more detail and have updated the conclusion to be more straightforward with our main findings. We have moreover made more clear in Section 2 what the claims we are attempting to confirm are and we now connect each subsection in Section 3 to these claims.
>
> Regarding study setups outside of Wang & Wang 22:
>
> We have made further experiments and added these to the updated version. These show (i.e. Table 2) that weighted contrast when there is a single dominating class tends to have similar positive areas to the original version but different negative areas. We also redid our experiments for ViT, now with a fine-tuned model on a fine-grained dataset (Food-101) in order to see how the contrastive method works when the differences between classes are smaller. These results are much clearer than in the original submission and show differences between the original methods and the weighted contrastive method, with the being more precise in its detail and being able to clearly show that the model takes details such as the noodles vs. rice in the pad thai and fried rice example in Figure 5b.
>
> We also add some further discussion under Section 3.3 regarding debugging the model using the contrastive weighted method and believe that it could be useful in finding hidden biases such as attending to the background of images rather than the main object.
>
> Thank you for your feedback and please do not hesitate to ask if there is anything else!
>
> Best regards,
> Authors

---

### Author Response · Authors · 2024-03-26

Dear Reviewers,

Thank you for all your valuable feedback. We have recently published a revised version which attempts to remedy much of the feedback that was given. Please do not hesitate to contact us regarding any further feedback or questions.

Best regards,
Authors

---

### Decision · Action_Editor_cmer · 2024-04-26

**Recommendation:** Reject

**Comment:**

This paper is a reproducibility report of the “Why Not Other Classes?”: Towards Class-Contrastive Back-Propagation Explanations (Wang & Wang, 2022).

Following the TMLR official acceptance criteria (https://jmlr.org/tmlr/acceptance-criteria.html), I mainly focused on whether this paper has sufficient generalized insights and actionable lessons for the TMLR audience.

As the reviewers' comments (especially for Reviewer h5Nh), this paper does not provide additional insights beyond the original paper. Furthermore, as the comment by Reviewer 5hNh, some claims in the paper are based on a few qualitative samples rather than more systemic and quantitative verification.

Likewise, I also think showing the extendability of the existing method to other architecture (e.g., ViT) or applying different attribute map extractors (XGradCAM and FullGrad) is not a sufficient contribution to bring new insight.

Overall, this paper does not provide sufficient generalized insights and actionable lessons for the TMLR audience. Therefore I recommend rejection.

**Audience:**

This paper has insufficient generalizable insights. As clarified in the TMLR official acceptance criteria (https://jmlr.org/tmlr/acceptance-criteria.html), it is unlikely to be of interested to the TMLR audience.

**Claims And Evidence:**

This paper aims to verify three claims of the original paper related to the weighted contrastive explanation. The experimental results show that these claims are consistent. In addition, this paper tests the generalizability of the original method to XGradCAM, FullGrad and ViT.